# PTH induces bone loss via microbial-dependent expansion of intestinal TNF$^+$ T cells and Th17 cells

Mingcan Yu[1,2], Abdul Malik Tyagi[1,2], Jau-Yi Li[1,2], Jonathan Adams[1,2], Timothy L. Denning[3], M. Neale Weitzmann[1,2,4], Rheinallt M. Jones[2,5,6] & Roberto Pacifici[1,2,6]*

Bone loss is a frequent but not universal complication of hyperparathyroidism. Using antibiotic-treated or germ-free mice, we show that parathyroid hormone (PTH) only caused bone loss in mice whose microbiota was enriched by the Th17 cell-inducing taxa segmented filamentous bacteria (SFB). SFB$^+$ microbiota enabled PTH to expand intestinal TNF$^+$ T and Th17 cells and increase their S1P-receptor-1 mediated egress from the intestine and recruitment to the bone marrow (BM) that causes bone loss. CXCR3-mediated TNF$^+$ T cell homing to the BM upregulated the Th17 chemoattractant CCL20, which recruited Th17 cells to the BM. This study reveals mechanisms for microbiota-mediated gut–bone crosstalk in mice models of hyperparathyroidism that may help predict its clinical course. Targeting the gut microbiota or T cell migration may represent therapeutic strategies for hyperparathyroidism.

[1] Division of Endocrinology, Metabolism and Lipids, Department of Medicine, Emory University, Atlanta, GA, USA. [2] Emory Microbiome Research Center, Emory University, Atlanta, GA, USA. [3] Center for Inflammation, Immunity & Infection, Institute for Biomedical Sciences, Georgia State University, Atlanta, GA, USA. [4] Atlanta VA Medical Center, Decatur, GA, USA. [5] Division of Pediatric Gastroenterology, Hepatology, and Nutrition, Department of Pediatrics, Emory University, Atlanta, GA, USA. [6] Immunology and Molecular Pathogenesis Program, Emory University, Atlanta, GA, USA. *email: roberto.pacifici@emory.edu

Parathyroid hormone (PTH) is a key calciotrophic hormone and a critical regulator of postnatal skeletal development[1]. Primary hyperparathyroidism, a condition characterized by chronic continuous overproduction of PTH by the parathyroid glands[2], is modeled in animals by continuous PTH (cPTH) infusion[3]. Primary hyperparathyroidism is a common cause of osteoporosis and fractures[4–6]. However, primary hyperparathyroidism is a heterogeneous disease as some patients develop bone loss while others do not. The presence or absence of hypercalcemia, with its absence referencing the normocalcemic variant, does not define or predict who will experience bone loss[2,7]. The underlying mechanism for this heterogeneity is unknown and no robust biomarker is available to predict the course of the disease. Continuous overproduction of PTH also occurs due to secondary causes such as chronic renal disease and vitamin D deficiency. The skeletal manifestations of secondary hyperparathyroidism are also heterogeneous[8,9]. This condition is modeled in animals by feeding a low calcium diet[10].

PTH signals by binding to the PTH-PTHrP receptor (PPR), which is expressed in cells of osteoblastic lineage[11–13], and T cells[14]. PTH may induce bone loss or promote bone mass acquisition, mainly depending on whether target cells are exposed to PTH continuously or intermittently. When produced or infused in excessive amounts and in a continuous fashion, PTH stimulates bone resorption and, to a lesser extent, bone formation leading to net bone loss[4,15].

cPTH fails to elicit bone waste in T cell null mice, or in mice with conditional deletion of PPR in T cells[16,17], implying that T cells participate in the mechanism of action of cPTH in bone. Direct targeting of T cells by PTH expands the population of BM TNF producing CD4+ and CD8+ T cells, and increases their production of TNF[17,18]. Another T cell population relevant for skeletal homeostasis are Th17 cells, a lineage of CD4+ cells defined by their capacity to produce the osteoclastogenic cytokine IL-17[19–21]. In the BM, IL-17 stimulates production of RANKL[18,22] and upregulates the RANKL receptor RANK[23], stimulating bone resorption[24]. In addition, IL-17 blunts bone formation by damping activation of Wnt signaling in osteoblasts[25,26]. Although in inflammatory conditions bone loss is induced by TNF and IL-17 released by BM T cells, many TNF producing T cells are generated in the small intestine in response to contact with common bacterial-released products, such as lipopolysaccharide (LPS) and flagellin[27]. In the mouse Th17 cells are also abundantly produced in the intestinal lamina propria in response to specific elements of the microbiota[28–30]. In healthy mice, intestinal Th17 cells are produced primarily in response to segmented filamentous bacteria (SFB)[28], which are spore-forming, Gram-positive commensal bacteria that potently induce differentiation of Th17 cells[29,30]. Infections with several extracellular pathogens such as Candida albicans and Citrobacter rodentium are also capable of activating Th17 cells.

It is presently unknown whether the T cells involved in the bone loss induced by PTH originate in the BM, or if they are first produced in the gut in response to the gut microbiota, and then home to the BM driven by PTH regulated mechanisms. Here we examined the role of the gut microbiota-PTH cross-talk in the generation of intestinal TNF+ T cells and Th17 cells, their homing to the BM, and their role in PTH-induced bone loss in mice. We found that cPTH treatment and low calcium diet do not induce bone loss in conventional mice treated with antibiotics or in germ-free (GF) mice, thus implicating the microbiome in the skeletal response to PTH. Moreover, we found that the presence of SFB within the intestinal microbiota is sufficient for PTH to exert its bone catabolic activity. We identify PTH-induced trafficking of TNF+ T cells and Th17 cells from the gut to the BM as a required pathway whereby PTH causes bone loss. Therefore, targeting the gut microbiota with antibiotics or blockade of T cell migration may represent therapeutic strategies for the treatment of hyperparathyroidism-induced bone loss.

## Results

**SFB+ microbiota is sufficient for PTH activity.** Recent studies have highlighted the importance of intestinal tissues and specific microbial taxa for the generation of Th17 cells[29,30]. To investigate the extent to which SFB influence PTH-induced bone loss in mice, C57BL/6 mice were purchased from a Taconic Biosciences vivarium that houses mice colonized with SFB (herein referred to as SFB+ TAC mice). In addition, C57BL/6 mice lacking SFB were purchased from The Jackson Laboratory (herein referred to as SFB−JAX mice). We also generated SFB+ JAX mice by fecal microbiome transfer (FMT) that involves oral gavaging SFB− JAX mice with a liquid suspension of fecal pellets collected from SFB+ TAC mice (Supplementary fig. 1a). SFB+ and SFB− female mice were infused with vehicle or cPTH for 2 weeks starting at 16 weeks of age, which is a treatment that models primary hyperparathyroidism[3,18,22]. A subset of mice was also treated with broad-spectrum antibiotics (1 mg/mL ampicillin, 0.5 mg/mL vancomycin, 1 mg/mL neomycin sulfate, 1 mg/mL metronidazole dissolved in water) for 4 weeks, starting at 14 weeks of age, to ablate the microbiota and thus assess the impact of the microbiome on the response to cPTH. Analysis of femurs collected at sacrifice by micro-computed tomography (μCT) revealed that in mice not treated with antibiotics (herein referred to as control mice), cPTH decreased trabecular bone volume fraction (BV/TV) and trabecular thickness (Tb.Th) in SFB+ TAC and SFB+ JAX mice, but not SFB− JAX mice (Fig. 1a–c). By contrast, cPTH failed to induce trabecular bone loss and alter trabecular structure in all groups of mice treated with antibiotics, indicating that SFB−containing microbiota was sufficient for cPTH to induce trabecular bone loss. Intriguingly, cortical bone area (Ct.Ar), total cross-sectional area inside the periosteal envelope (Tt.Ar), and average cortical thickness (Ct.Th), which are indices of cortical structure, were significantly decreased by cPTH in all groups of mice regardless of antibiotic treatment (Supplementary Fig. 2a–c), suggesting that cPTH caused cortical bone loss via a microbiome-independent mechanism.

PTH induces bone loss by stimulating bone resorption, a phenomenon partially mitigated by a compensatory increase in bone formation. Histomorphometric analysis of femoral cancellous bone demonstrated that cPTH increased two indices of bone resorption -osteoclast number (N.Oc/BS) and osteoclast surfaces (Oc.S/BS)- in SFB+ TAC and SFB+ JAX mice, but not in either SFB− JAX mice or all groups of antibiotic-treated mice (Fig. 1d, f, g). cPTH also increased two dynamic indices of bone formation—mineral apposition rate (MAR) and bone formation rate (BFR/BS)—in SFB+ mice. By contrast, cPTH did not alter bone formation in SFB− JAX mice or all groups of antibiotic-treated mice (Fig. 1e, h, i). Measurements of serum levels of CTX and osteocalcin, which are markers for bone resorption and bone formation respectively, revealed that cPTH increased CTX and osteocalcin levels in control SFB+ TAC and SFB+ JAX mice, while it did not increase CTX and osteocalcin levels in SFB− JAX mice and all groups of antibiotic-treated mice (Fig. 1j, k).

To investigate the role of SFB in the bone loss associated with secondary hyperparathyroidism continuous endogenous overproduction of PTH was induced by feeding mice a low calcium diet (0.01% calcium) for 4 weeks, starting at 16 weeks of age (Supplementary Fig. 1b). This strategy models secondary hyperparathyroidism and induces bone loss in a manner similar to cPTH[13]. A diet containing 1.75 % calcium was used as control diet. In mice not treated with antibiotics, a low calcium diet

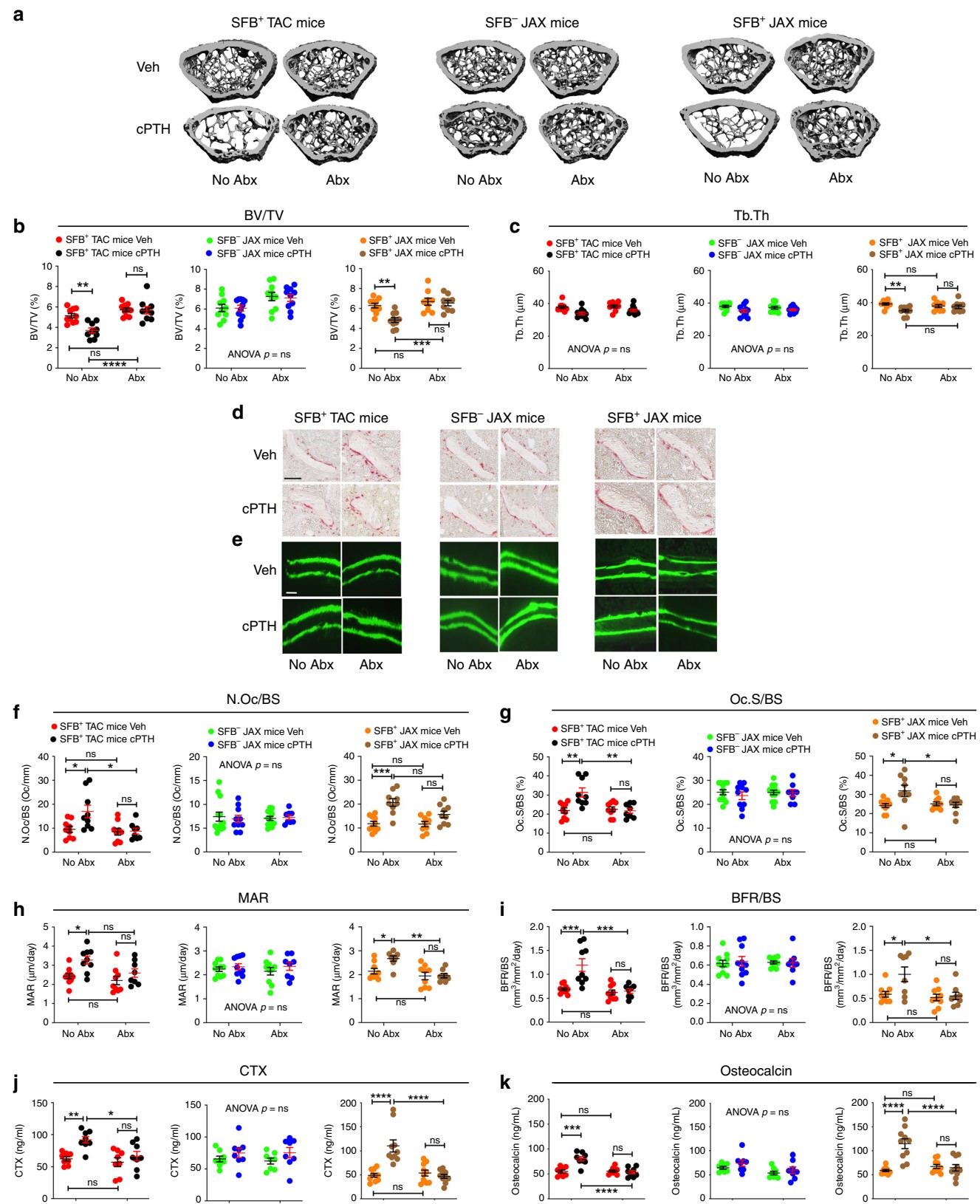

lowered BV/TV and Tb.Th in SFB+ JAX and SFB+ TAC mice. By contrast, a low calcium diet resulted in no changes in femoral BV/TV and Tb.Th in SFB− JAX mice, or in all groups of mice treated with antibiotics (Fig. 2a–c). Moreover low calcium diet increased CTX and osteocalcin levels in control SFB+ mice, but did not do so in control SFB− JAX mice and all groups of mice treated with antibiotics (Fig. 2d, e). These data indicated that colonization with SFB was sufficient for PTH to stimulate bone turnover and induce trabecular bone loss in models of both primary and secondary hyperparathyroidism.

**Fig. 1 SFB$^+$ microbiota is sufficient for cPTH to induce trabecular bone loss and stimulate bone turnover.** SFB$^+$ and SFB$^-$ mice from Taconic (TAC) and Jackson Laboratory (JAX) were treated with vehicle or cPTH for 2 weeks with antibiotics (Abx) or without antibiotics (No Abx). **a** The figure shows images of representative 3-dimensional μCT reconstructions of examined femurs. **b** Femoral trabecular bone volume fraction (BV/TV), $n = 9$–11 mice per group. **c**. Trabecular thickness (Tb.Th), $n = 9$–11 mice per group. **d** The images show representative tartrate resistant acid phosphatase (TRAP) stained sections of the distal femur. Original magnification ×40. Scale bar represents 300 μm. **e** Images are representative sections displaying the calcein double-fluorescence labeling. Original magnification ×20. Scale bar represents 300 μm **f** Number of osteoclasts per mm bone surface (N.Oc/BS), $n = 8$–10 mice per group. **g** Percentage of bone surface covered by osteoclasts (Oc.S/BS), $n = 8$–10 mice per group. **h** Mineral apposition rate (MAR), $n = 8$–10 mice per group. **i** Bone formation rate per mm bone surface (BFR/BS), $n = 8$–10 mice per group. **j** Serum levels of CTX, a marker of bone resorption, $n = 8$–10 mice per group. **k** Serum levels of osteocalcin, a marker of bone formation, $n = 8$–10 mice per group. All data are expressed as Mean ± SEM. All data were normally distributed according to the Shapiro-Wilk normality test. Data were analyzed by two-way ANOVA and post hoc tests applying the Bonferroni correction for multiple comparisons. *$p < 0.05$, **$p < 0.01$, ***$p < 0.001$, and ****$p < 0.0001$ compared to the indicated group. Source data are provided as a Source Data file.

To investigate whether SFB is specifically required and sufficient for PTH-induced bone loss, GF mice were subjected to FMTs to generate mice colonized exclusively with SFB (herein referred to as SFB mono-associated mice) (Supplementary fig. 1c and Supplementary fig. 3). Controls included GF mice colonized with the SFB$^-$ JAX microbiota and mice colonized with both SFB and the SFB$^-$ JAX microbiota. Due to the technical challenge of carrying out surgical procedures in the GF mice housing cages, hyperparathyroidism was induced only by feeding a low calcium diet for 4 weeks. Low calcium diet administration resulted in no loss of femoral BV/TV and Tb.Th, and no increases in CTX and osteocalcin levels in GF mice, in SFB mono-associated mice, or in GF mice colonized with SFB$^-$ JAX microbiota (Fig. 3a–e). By contrast, low calcium diet lowered BV/TV, altered trabecular structure and increased CTX and osteocalcin levels in mice colonized with both SFB and the SFB$^-$ JAX microbiota (Fig. 3a–e). These data revealed that intestinal colonization with SFB alone was not sufficient for PTH to stimulate bone turnover and induce trabecular bone loss.

**SFB$^+$ microbiota is sufficient for PTH to expand Th17 cells.**
We used flow cytometry to assess the effect of cPTH and low calcium diet on the number of Peyer's patches (PP) and BM Th17 cells (Supplementary Fig. 4). Since the calculation of the absolute number of PP Th17 cells is technically challenging due to variability of the size of the collected PP tissue, PP Th17 cells were quantified only as percentage of total CD4$^+$ T cells. This analysis revealed that in mice not treated with antibiotics both cPTH and low calcium diet increased the number of PP and BM Th17 cells in SFB$^+$ TAC and SFB$^+$ JAX mice, but not in SFB$^-$ JAX mice or in all groups of mice treated with antibiotics (Figs. 4a, b and 5a, b). BM Th17 cells are relevant for the activity of PTH as they produce the osteoclastogenic factor IL-17A. Accordingly, cPTH or low calcium diet increased *Il17a* transcripts in in the BM and the small intestine (SI) in SFB$^+$ TAC and SFB$^+$ JAX mice, but not in SFB$^-$ JAX mice, or in all groups of antibiotic-treated mice (Figs. 4c, d and 5c, d).

Further experiments revealed that a low calcium diet increased the number of PP and BM Th17 cells in mice colonized with both SFB and the SFB$^-$ JAX microbiota, while it failed to increase intestinal and BM Th17 cells in GF mice, in SFB mono-associated mice, and in GF mice colonized with the SFB$^-$ JAX microbiota but not SFB (Fig. 5e, f). These findings demonstrated that the presence of SFB within the gut microbiota is sufficient for cPTH to increase Th17 cells and their production of IL-17A.

Since IL-17A is also produced by γδ T cells and neutrophils, we also examined the effects of cPTH on these cells. cPTH increased the relative frequency of PP IL-17A$^+$ CD4 + T cells (Th17 cells) but not of PP IL-17A$^+$ γδ T cells (Supplementary fig. 5a). The frequency of neutrophils in PPs was too low to reliably assess

their production of IL-17. Moreover, cPTH did not affect the frequency of PP neutrophils (Supplementary fig. 5b).

**SFB$^-$ microbiota is sufficient for PTH to expand TNF$^+$ T cells.**
IL-1β and IL-6 are cytokines that are required for the generation of intestinal Th17 cells by partnering with TGFβ. We found that cPTH increased the SI transcript levels of *Tgfb*, but not those of *Il1b* and *Il6* (Supplementary Fig. 6), suggesting that alternative factors may be implicated along with TGFβ in the expansion of Th17 cells induced by PTH. A cytokine critical for the capacity of PTH to induce the expansion of BM Th17 cells is TNF[18], which is produced by BM CD4$^+$ and CD8$^+$ T cells in response to direct targeting of T cells by PTH[17]. However, the effects of cPTH and low calcium diet on intestinal TNF production are unknown. In keeping with reports that common bacterial components (e.g. LPS and flagellin) induce T cell activation and T cell TNF production in the gut[27], we found that cPTH or low calcium diet increased the number of PP and BM TNF$^+$ T cells in both SFB$^+$ and SFB$^-$mice, but did not expand the number of TNF$^+$ T cells in all groups of antibiotic-treated mice (Figs. 6a, b and 7a, b). Moreover, cPTH or low calcium diet increased *Tnf* transcripts in the SI and BM of both SFB$^+$ and SFB$^-$ mice, but not in the SI and BM of antibiotic-treated mice (Figs. 6c, d and 7c, d).

Further studies showed that a low calcium diet increased the number of intestinal and BM TNF$^+$ T cells in both GF mice colonized with the microbiota of SFB$^-$ JAX mice or in mice colonized with the SFB$^-$JAX microbiota and SFB, while it failed to increase intestinal and BM TNF$^+$ T cells in GF mice or in SFB monoassociated mice (Fig. 7e, f). These findings indicated that cPTH expanded the population of TNF$^+$ T cells in the gut and the BM via a microbiota-dependent, but SFB$^-$independent mechanism.

**TNF mediates migration of intestinal Th17 cells to the BM.**
SFB induces Th17 cell expansion in the gut with focused antigenic specificity. As a result, most of the Th17 cells produced in the lamina propria in response to SFB contain the vβ14 chain in their TCR receptor[31,32]. In SFB$^+$ TAC and SFB$^+$ JAX mice, cPTH or a low calcium diet increased the number vβ14$^+$ Th17 cells in the BM, which was a phenomenon that was blocked by antibiotics (Fig. 8a). Similarly, a low calcium diet increased the number of BM Vβ14$^+$ Th17 cells in mice colonized with both SFB and the SFB$^-$ JAX microbiota, while it failed to expand the population of BM Vβ14$^+$ Th17 cells in GF mice, in SFB mono-associated mice, or in mice colonized with SFB$^-$ JAX microbiota only (Fig. 8b). Together, these findings indicated that the expansion of BM Th17 cells induced by PTH was related to migration of intestinal Th17 cells to the BM. The effects of cPTH and a low calcium diet on helper T cell differentiation were limited to Th17 cells, as we did

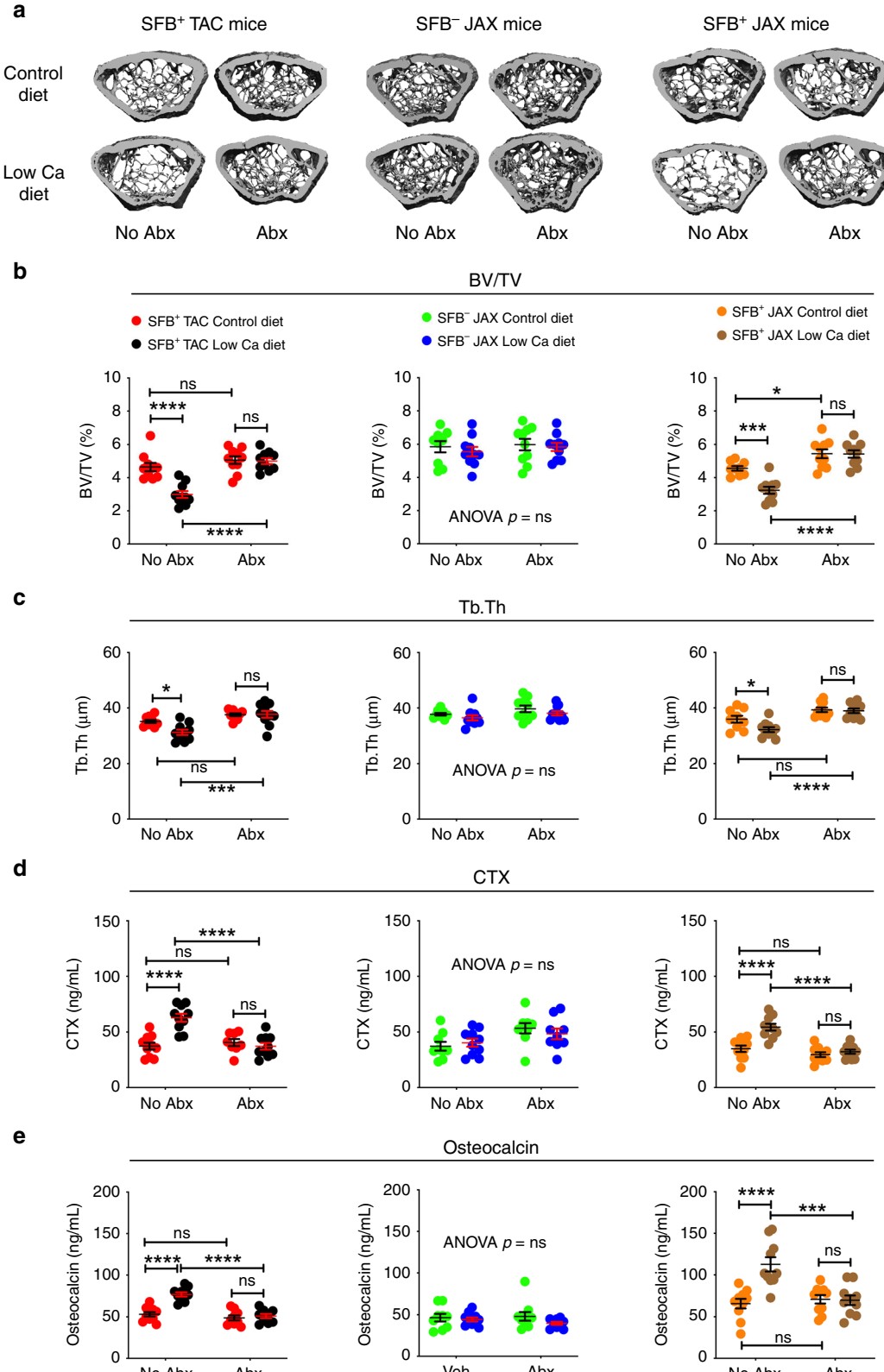

**Fig. 2 SFB+ microbiota is sufficient for low calcium diet to induce trabecular bone loss and stimulate bone turnover.** SFB+ TAC and JAX mice and SFB- JAX mice were treated with control diet or low calcium diet for 4 weeks and with antibiotics (Abx) or without antibiotics (No Abx). **a** The figure shows images of representative three-dimensional μCT reconstructions of examined femurs. **b** Femoral trabecular bone volume/total volume fraction (BV/TV), $n = 9$–10 mice per group. **c** Trabecular thickness (Tb.Th), $n = 9$–10 mice per group. **d** Serum levels of CTX, a marker of bone resorption, $n = 8$–10 mice per group. **e** Serum levels of osteocalcin, a marker of bone formation, $n = 9$–10 mice per group. All data are expressed as Mean ± SEM. All data were normally distributed according to the Shapiro-Wilk normality test. Data were analyzed by two-way ANOVA and post hoc tests applying the Bonferroni correction for multiple comparisons. $*p < 0.05$, $***p < 0.001$, and $****p < 0.0001$ compared to the indicated group. Source data are provided as a Source Data file.

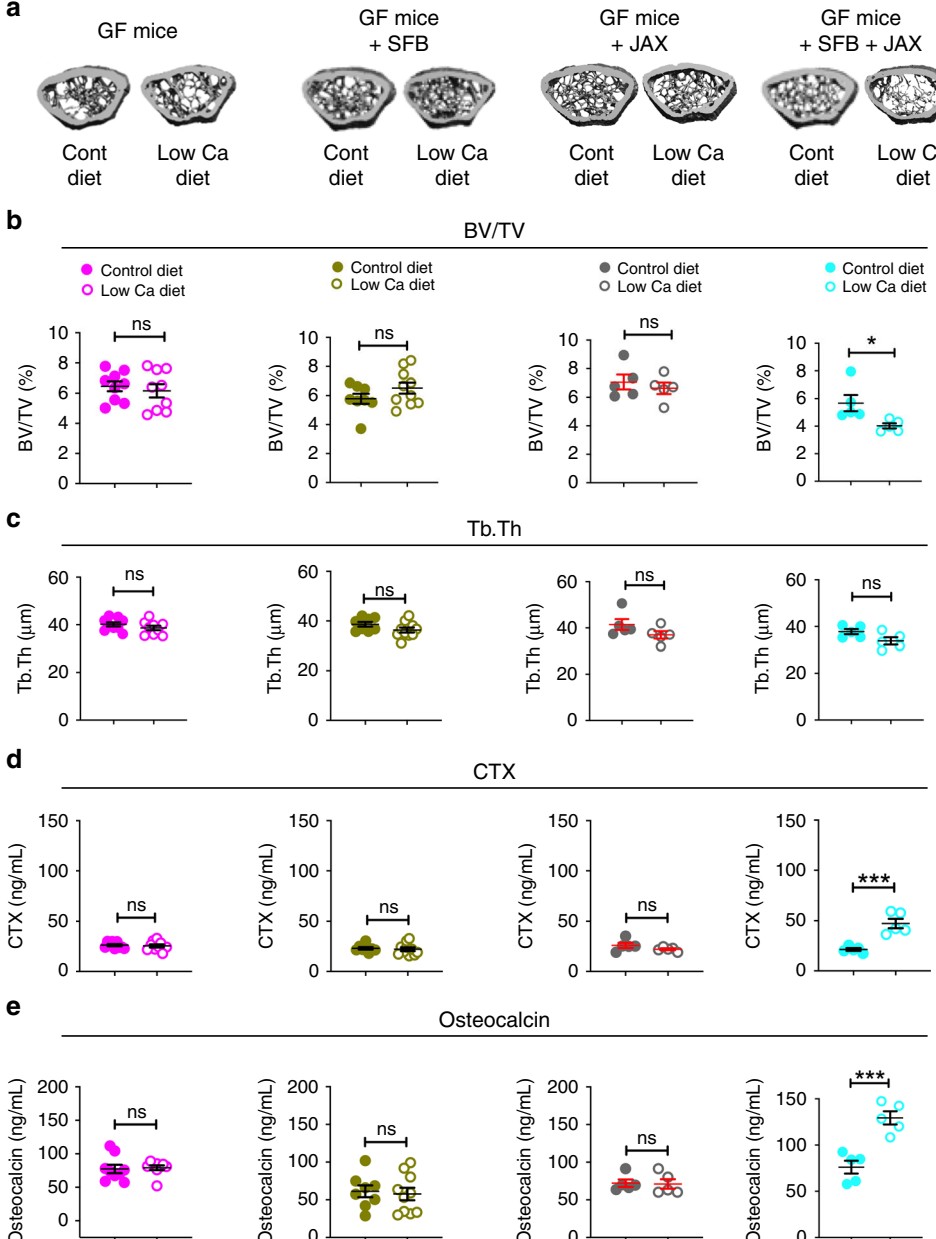

**Fig. 3 SFB⁺ microbiota is sufficient for low calcium diet to alter bone structure and turnover in GF mice subjected to fecal material transfer.** Germ-free (GF) mice, SFB monoassociated GF mice, and GF mice colonized with JAX mice microbiota ±SFB were treated with control diet or low calcium diet for 4 weeks. **a** The figure shows images of representative three-dimensional μCT reconstructions of examined femurs. **b** Femoral trabecular bone volume/total volume fraction (BV/TV), n = 5–10 mice per group. **c** Trabecular thickness (Tb.Th), n = 5–10 mice per group. **d** Serum levels of CTX, a marker of bone resorption, n = 5–10 mice per group. **e** Serum levels of osteocalcin, a marker of bone formation, n = 5–10 mice per group. Data are expressed as Mean ± SEM. All data were normally distributed according to the Shapiro-Wilk normality test and analyzed by unpaired t-tests. *p < 0.05 and ***p < 0.001 compared to the indicated group. Source data are provided as a Source Data file.

not detect changes in the population of regulatory T cells, IFNγ⁺ CD4 + T cells, or in IL-4⁺ CD4 + T cells in the small intestine or the BM of SFB⁺JAX mice in response to cPTH and a low calcium diet (Supplementary fig. 7a, b).

IL-17A-eGFP mice were then used to confirm that cPTH triggers the homing of Th17 cells from the gut to the BM. In addition, SFB⁻ JAX Tnf−/− mice were used to test the specific role of TNF in driving the migration of intestinal Th17 cells to the BM. The IL-17A-eGFP reporter mice possess an IRES-eGFP sequence after the stop codon of the Il17a gene, so that eGFP expression is limited to IL-17A expressing cells, thus allowing Th17 cells to be detected by measuring eGFP by flow cytometry.

Splenic naïve CD4⁺ cells (CD4⁺CD44ˡᵒCD62Lʰⁱ cells) were purified from IL-17A-eGFP mice and cultured in Th17 cell-polarizing conditions for 4 days. Th17 cells (CD4⁺eGFP⁺ cells) were then FACS sorted and transferred into WT and Tnf −/− SFB⁻ JAX mice that had been treated with vehicle or cPTH for 6 days. After 3 days of further vehicle or cPTH treatment, the recipient mice were sacrificed and BM Th17 cells (CD4⁺eGFP⁺ cells) counted. WT SFB⁻ JAX mice treated with cPTH had a higher number of BM Th17 cells than those treated with vehicle (Fig. 8c, d). By contrast, cPTH did not increase the relative and absolute frequency of BM Th17 cells in SFB⁻ JAX Tnf−/− mice. These findings demonstrated that cPTH increases the recruitment

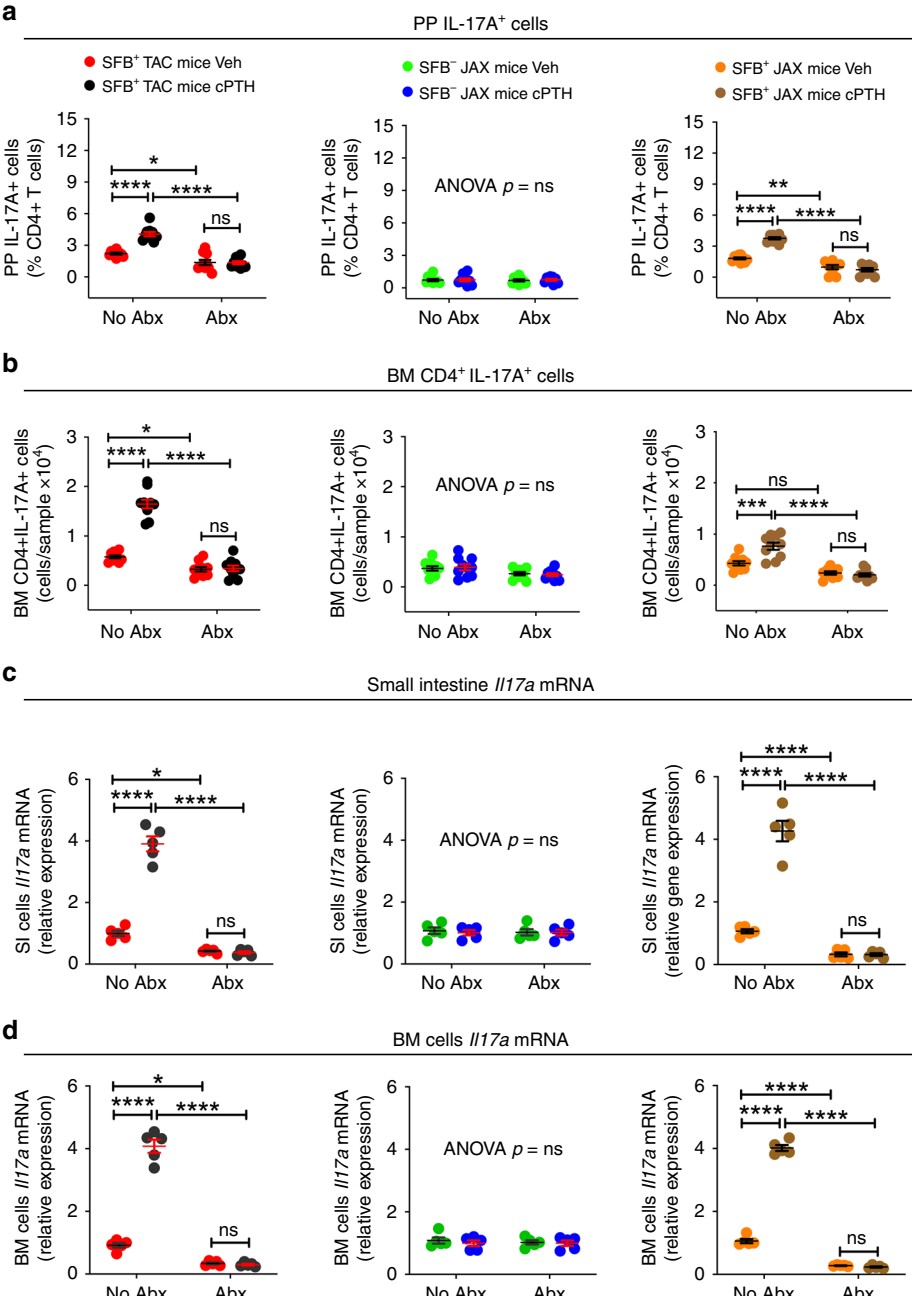

**Fig. 4 SFB$^+$ microbiota is sufficient for cPTH to expand intestinal and BM Th17 cells. a** Relative frequency of PP Th17 cells, $n = 9$–10 mice per group. **b** Absolute number of BM Th17 cells, $n = 9$–10 mice per group. **c** Small Intestine levels of *Il17a* transcripts, $n = 5$ mice per group. **d** BM levels of *Il17a* transcripts, $n = 5$ mice per group. SFB$^+$ TAC and JAX mice and SFB$^-$ JAX mice were treated with cPTH with antibiotics (Abx) or without antibiotics (No Abx) for 2 weeks. Data are expressed as Mean ± SEM. All data were normally distributed according to the Shapiro-Wilk normality test. Data were analyzed by two-way ANOVA and post hoc tests applying the Bonferroni correction for multiple comparisons. *$p < 0.05$, **$p < 0.01$,***$p < 0.001$, and ****$p < 0.0001$ compared to the indicated group. Source data are provided as a Source Data file.

of Th17 cells to the BM via a TNF-dependent, but SFB −independent mechanism.

Th17 cells are generated in the intestinal lamina propria from where they migrate to PPs and then to distant organs driven by chemokine gradients[33,34]. Th17 cells express the chemokine receptor CCR6[35], which binds to its ligand CCL20. We found that cPTH increased *Ccl20* mRNA expression in BM cells in both SFB$^+$ and SFB$^-$ C57BL/6 mice, but not in the BM of corresponding groups of antibiotic-treated mice (Fig. 8e). Analysis of purified BM cell populations revealed that cPTH increased *Ccl20* expression in stromal cells but not in T cells, B cells, monocytes

or dendritic cells (Fig. 8e). CCL20 is strongly induced by inflammatory cytokines[33]. This suggests that the increased production of TNF in the BM induced by cPTH in both SFB$^+$ and SFB$^-$ mice may upregulate CCL20 expression in BM cells, causing the chemotactic migration of Th17 cells from the gut to the BM. To investigate the role of TNF in Th17 cell migration in the context of a Th17 cell-inducing microbiome, we utilized SFB$^+$ JAX WT and *Tnf*−/− mice, and found that cPTH expanded PP Th17 cells, BM Th17 cells, and BM Vβ14$^+$ Th17 cells in WT mice but not in *Tnf*−/− mice (Fig. 8g). Importantly, cPTH upregulated BM cell expression of *Ccl20* in WT SFB$^+$ JAX mice, but not in

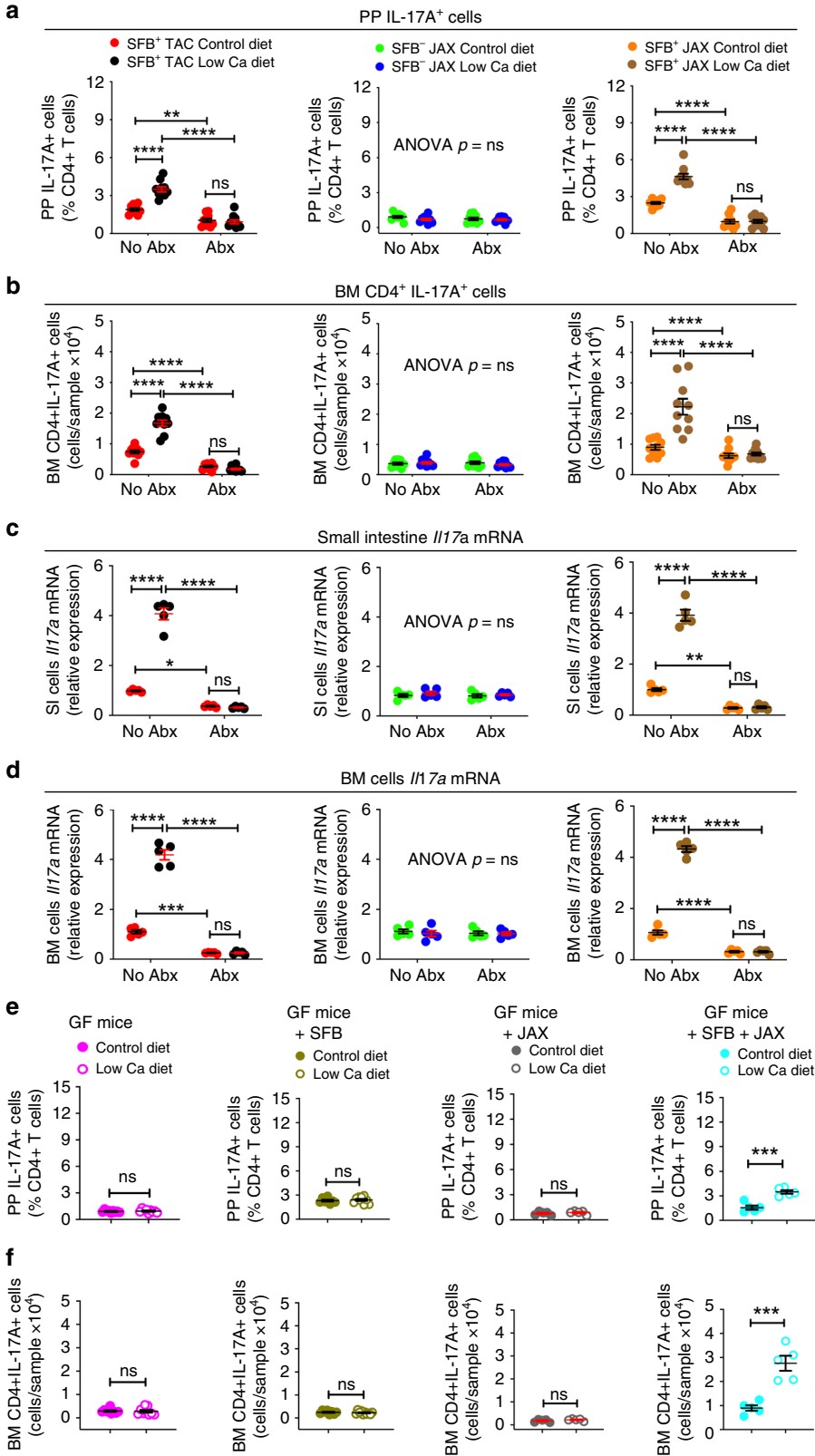

*Tnf*−/− SFB+ JAX mice (Fig. 8h). Next, splenic T cells from WT and *Tnf*−/− mice were transferred into *Tcrb*−/− mice, a strain lacking αβ T cells. After 2 weeks, a length of time sufficient for the engraftment and expansion of donor T cells, recipient mice were treated with vehicle or cPTH for 2 weeks. cPTH increased expression of *Ccl20* in the BM of host mice with WT T cells but not in those with *Tnf*−/− T cells (Fig. 8i). These findings demonstrate that the production of TNF by T cells is required for cPTH to upregulate *Ccl20* expression. Together these finding indicated that TNF did not directly induce the differentiation and expansion of Th17 cells in the BM, but rather, expanded intestinal Th17 cells in concert with SFB, and directed the migration of

**Fig. 5 SFB+ microbiota is sufficient for low calcium diet to expand intestinal and BM Th17 cells. a–d** Frequency of PP and BM Th17 cells and small intestine (SI) and BM levels of *Il17a* transcripts in SFB+ TAC and JAX and SFB- JAX mice treated with control diet or low calcium diet for 4 weeks with antibiotics (Abx) or without antibiotics (No Abx), **a**, **b** $n = 9$–10 mice per group; **c**, **d** $n = 5$ mice per group. **e**, **f** Frequency of PP and BM Th17 cells in GF mice, SFB monoassociated GF mice, and GF mice colonized with JAX mice microbiota ±SFB treated with control diet or low calcium diet for 4 weeks, $n = 5$–10 mice per group. Data are expressed as Mean ± SEM. All data were normally distributed according to the Shapiro-Wilk normality test. Data in **a–d** were analyzed by two-way analysis-of-variance and post hoc tests applying the Bonferroni correction for multiple comparisons. Data in panels e,f were analyzed by unpaired *t*-tests. *$p < 0.05$, **$p < 0.01$, ***$p < 0.001$, and ****$p < 0.0001$ compared to the indicated group. Source data are provided as a Source Data file.

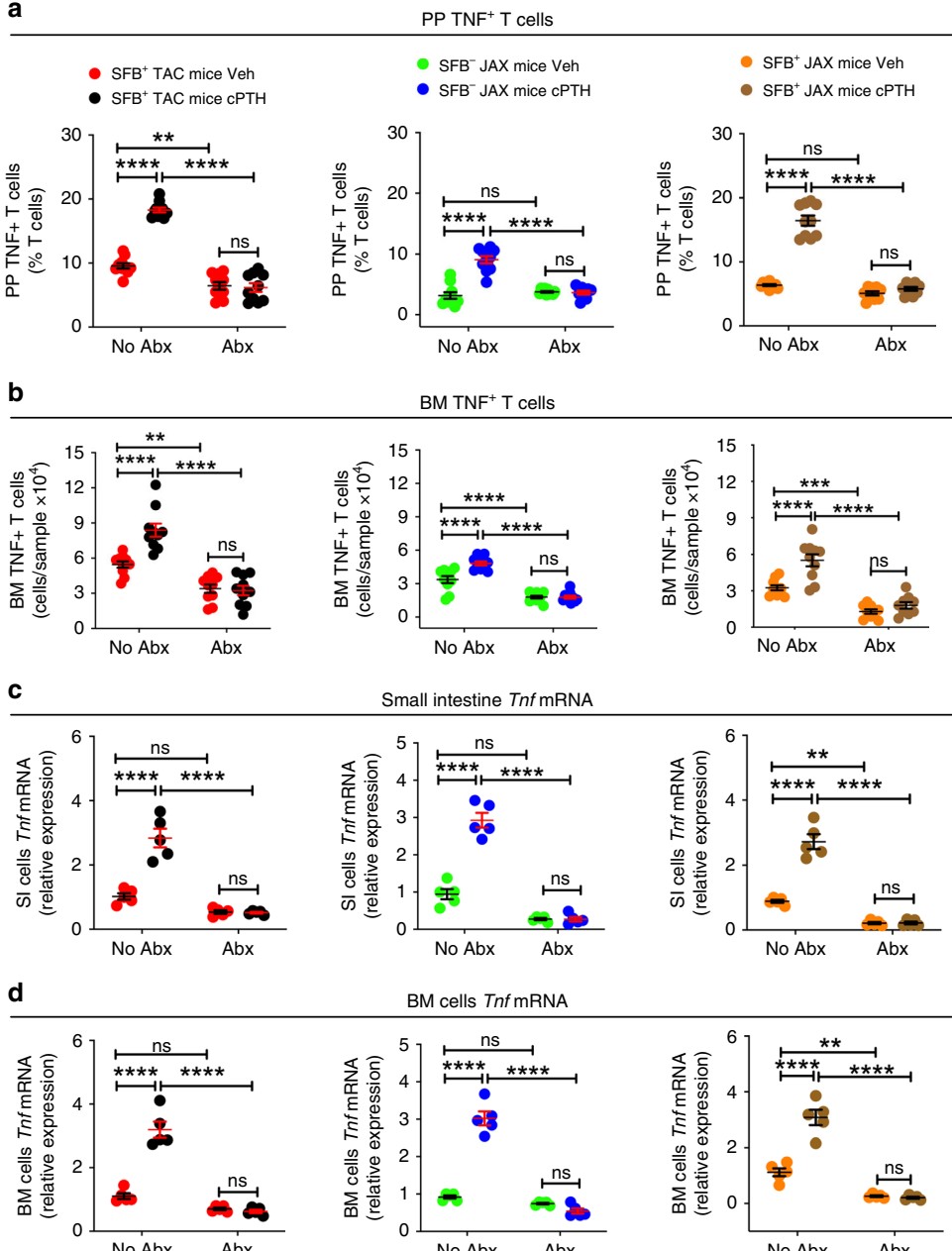

**Fig. 6 SFB− microbiota is sufficient for cPTH to expand intestinal and BM TNF+ T cells. a** Relative frequency of PP TNF+ T cells, $n = 9$–10 mice per group. **b** Absolute number of BM TNF+ T cells, $n = 9$–10 mice per group. **c**, **d** SI and BM levels of *Tnf* transcripts. SFB+ TAC and JAX mice and SFB- JAX mice were treated with cPTH with antibiotics (Abx) or without antibiotics (No Abx) for 2 weeks, $n = 5$ mice per group. All data were normally distributed according to the Shapiro-Wilk normality test. Data were analyzed by two-way ANOVA and post hoc tests applying the Bonferroni correction for multiple comparisons. **$p < 0.01$, ***$p < 0.001$, and ****$p < 0.0001$ compared to the indicated group. Source data are provided as a Source Data file.

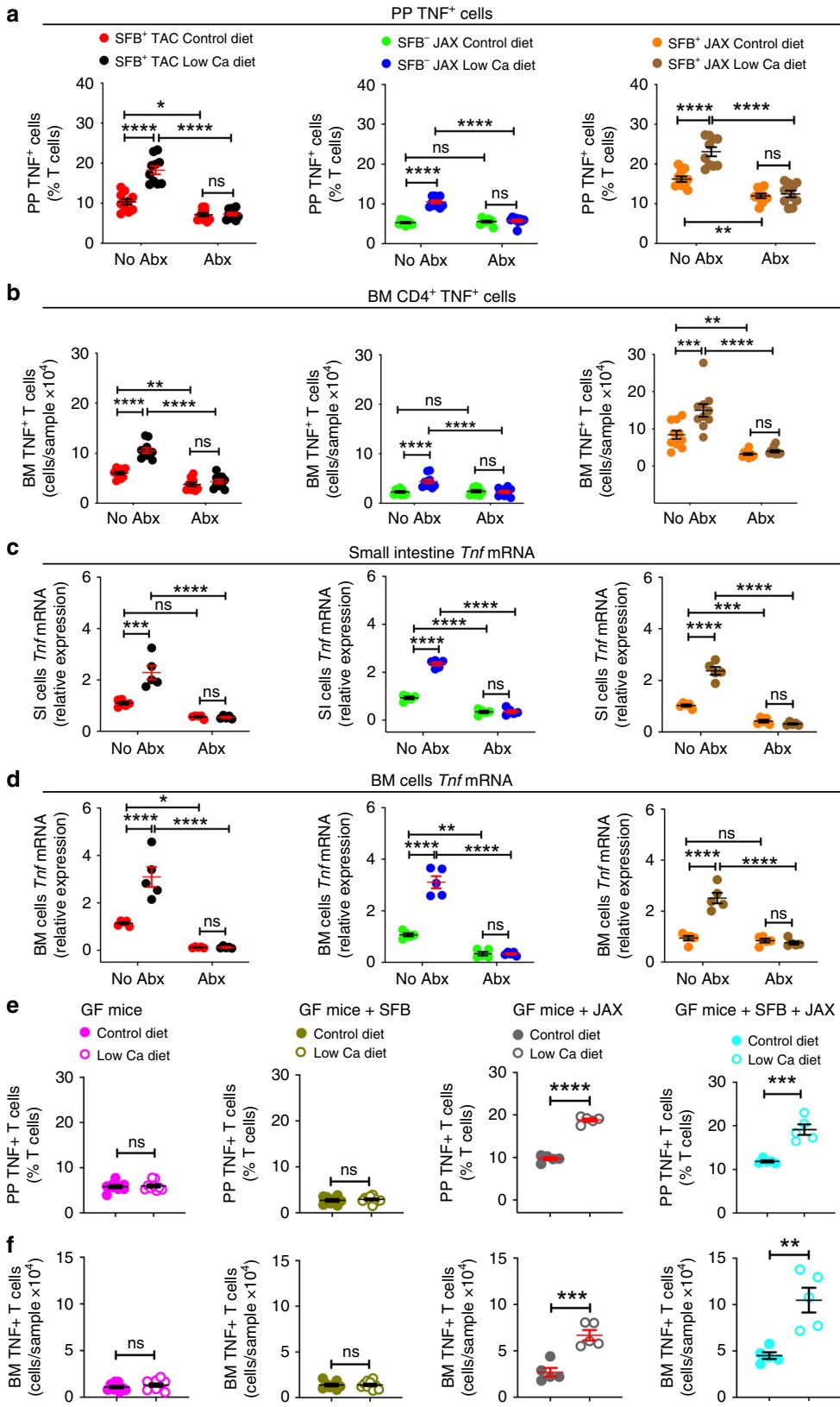

intestinal Th17 cells to the BM by upregulating CCL20. Confirming the functional relevance of TNF for the bone catabolic activity of PTH, cPTH also failed to decrease BV/TV and increase the levels of CTX and osteocalcin in *Tnf−/−* SFB+ JAX mice (Fig. 8j).

**S1PR1 induces T cell egress from the small intestine.** T cells express sphingosine 1 phosphate (S1P) receptor 1 (SIP1R1), which promotes lymphocyte egress from lymphoid organs in response to sensing of circulating S1P[36]. This suggest that PTH may promote the egress of TNF+ T cells and Th17 cells from the

**Fig. 7 SFB⁻ microbiota is sufficient for low calcium diet to expand intestinal and BM TNF⁺ T cells. a–d** Frequency of PP and BM TNF⁺ T cells and small intestine (SI) and BM levels of *Tnf* transcripts in SFB⁺ TAC and JAX and SFB⁻ JAX mice treated with control diet or low calcium diet for 4 weeks with antibiotics (Abx) or without antibiotics (No Abx), **a**, **b** $n = 9$–10 mice per group; **c**, **d** $n = 5$ mice per group. **e**, **f**. Frequency of PP and BM TNF⁺ T cells in GF mice, SFB monoassociated GF mice, and GF mice colonized with JAX mice microbiota ±SFB treated with control diet or low calcium diet for 4 weeks, $n = 5$–10 mice per group. Data are expressed as Mean ± SEM. All data were normally distributed according to the Shapiro-Wilk normality test. Data in **a**–**d** were analyzed by two-way analysis-of-variance and post hoc tests applying the Bonferroni correction for multiple comparisons. Data in **e**, **f** were analyzed by unpaired *t*-tests. *$p < 0.05$, **$p < 0.01$, ***$p < 0.001$, and ****$p < 0.0001$ compared to the indicated group. Source data are provided as a Source Data file.

intestine through a SIP1R1 mediated mechanism. To determine if PTH promotes the exit of PP TNF⁺ T cells and Th17 from the intestine and does so via SIP1R1, SFB⁺ TAC mice were treated with cPTH and the S1PR1 functional antagonist FTY720, which is an agent that arrests lymphocyte exit from PPs and mesenteric lymph nodes without affecting lymphocyte function[37,38]. FTY720 did not alter the capacity of cPTH to increase PP Th17 cells, but it blocked the increase in BM Th17 cells and BM Vβ14⁺ Th17 cells induced by cPTH (Fig. 9a). Analysis of control mice treated with PTH-vehicle revealed that in this group FTY720 did not decrease the frequency of PP Th17 cells but did decrease the number of BM Th17 cells and BM Vβ14⁺ Th17 cells (Fig. 9a). Similarly, FTY720 did not block the increase in PP TNF⁺ T cells induced by cPTH, but it prevented the increase in BM TNF⁺ T cells induced by cPTH (Fig. 9b). In addition, FTY720 lowered the number of BM TNF⁺ T cells but not the frequency of PP TNF⁺ T cells in PTH-vehicle treated mice (Fig. 9b). Attesting to the functional relevance of these effects, FTY720 completely prevented the loss of BV/TV and Tb.Th and the increase in serum CTX and osteocalcin induced by cPTH (Fig. 9c). Together, these findings showed that the egress of TNF⁺ T cells and Th17 cells from the gut and their homing to the BM is mediated by S1PR1 signaling.

**CCL20 and CXCR3 guide the influx T cell into the BM.** Following their exit from the intestine, Th17 cells migrate to sites of inflammation guided by the CCR6/CCL20 axis[39]. cPTH upregulated *Ccl20* in the BM (Fig. 8e, f), providing a mechanism for PTH to attract Th17 cells to the BM. To determine the role of CCL20 driven influx of Th17 cells into the BM for the mechanism of action of cPTH in bone, SFB⁺ TAC mice were treated with cPTH and a neutralizing anti-CCL20 antibody or isotype matched irrelevant antibody. CCL20 antibody did not alter the capacity of cPTH to increase PP Th17 cells but prevented the increase in BM Th17 cells and Vβ14⁺ Th17 cells induced by cPTH (Fig. 9d). Furthermore, CCL20 antibody did not block the increase in PP TNF⁺ T cells induced by cPTH (Fig. 9e), but partially blocked the increase in BM TNF⁺ T cells induced by cPTH, which indicated that CCL20 may contribute to regulate the influx of TNF⁺ T cells into the BM. Supporting evidence for an essential function of CCL20, CCL20 antibody also blocked the loss of BV/TV and Tb.Th and the increase in serum CTX and osteocalcin induced by cPTH (Fig. 9f).

The homing of some cytokine-producing T cells to lymphoid organs is dependent on the expression of *Cxcr3* on T cells[40]. To determine if CXCR3 was required for PTH to attract TNF⁺ T cells to the BM, SFB⁺JAX *Cxcr3*−/− mice and WT littermates were treated with cPTH for 2 weeks and then analyzed. These experiments were conducted using SFB⁺ mice to ascertain whether the recruitment of TNF⁺ T cells to the BM is required for the recruitment of Th17 cells to the BM. cPTH increased PP Th17 cells, BM Th17 cells, and BM Vβ14⁺ T cells in SFB⁺ WT but not in SFB⁺ *Cxcr3*−/− mice (Fig. 9g). Similarly, cPTH increased PP and BM TNF⁺ T cells in SFB⁺ WT but not in SFB⁺ *Cxcr3*−/− mice (Fig. 9h). In addition, cPTH decreased BV/TV and Tb.Th and increased the levels of CTX and osteocalcin in SFB

⁺ WT but not in SFB⁺ *Cxcr3*−/− mice (Fig. 9i). Altogether, the data were consistent with the hypothesis that the migration of TNF⁺ T cells to the BM is required for cPTH to induce the subsequent homing of intestinal Th17 cells to the BM and to induce bone loss.

**Discussion**
We reported that in mice models of primary and secondary hyperparathyroidism, the presence of SFB in the gut microbiota, and the SFB-dependent expansion of intestinal Th17 cells were sufficient for PTH to stimulate bone resorption and induce bone loss. We also showed that the microbiota was required for PTH to expand intestinal TNF producing T cells, which in concert with SFB, induced the expansion of Th17 cells. While expansion of intestinal and BM TNF⁺ T cells alone was not sufficient for PTH to exert its bone catabolic activity, intestinal TNF producing T cells were required for PTH to increase the number of intestinal Th17 cells in PP. Migration of TNF⁺ T cells to the BM and production of TNF by T cells were required for upregulating the expression of CCL20 on BM stromal cells, and for the homing of Th17 cells to the BM. We found that TNF⁺ T cells and Th17 cells egressed the intestine through a S1PR1 dependent mechanism, and the influx of TNF⁺ T cells and Th17 cells into the BM to be guided by CXCR3 and CCL20.

Th17 cells play a pivotal role in the bone loss resulting from pathologic conditions such as psoriasis, rheumatoid arthritis, periodontal disease, and inflammatory bowel disease[20,41]. In these inflammatory states, Th17 cells potently induce osteoclastogenesis by secreting IL-17A, RANKL, TNF, IL-1, and IL-6, along with low levels of IFNγ[42–44]. While Th17 cells contribute to pathologic bone destruction in autoimmune diseases, the bone erosion induced by Th17 cells serves an important protective role in diseases such as periodontitis, which is one of the most common infectious diseases, where Th17 cell-induced tooth loss is critical for eradicating bacteria from infected oral cavities and surfaces[45].

In the current study, we found that PTH induced the expansion of intestinal TNF⁺ T cells and Th17 cells, which then migrated to the BM. While intestinal TNF⁺ T cells were essential for the expansion of Th17 cells in the gut, BM TNF⁺ T cells were required to upregulate CCL20 and chemoattract Th17 cells to the BM. The latter population was ultimately responsible for bone loss, via secretion of IL-17A. Stimulation of intestinal Th17 cell differentiation by PTH was dependent on the presence of SFB. However, SFB required the concomitant stimulation from additional elements of the microbiota to induce bone loss as SFB alone was not capable of expanding intestinal TNF⁺ T cells, which are formed in response to commonly released bacterial products such as LPS. PTH also increased the intestinal production of TGFβ, while it did not alter the local production of IL-6 or IL-1β, which are required for the canonical pathway of Th17 cell differentiation[46,47]. These findings suggest a complex effect of PTH on intestinal Th17 cell differentiation involving both TGFβ and TNF, where TNF is likely to act as a partner of TGFβ, in lieu of IL-6 or IL-1β.

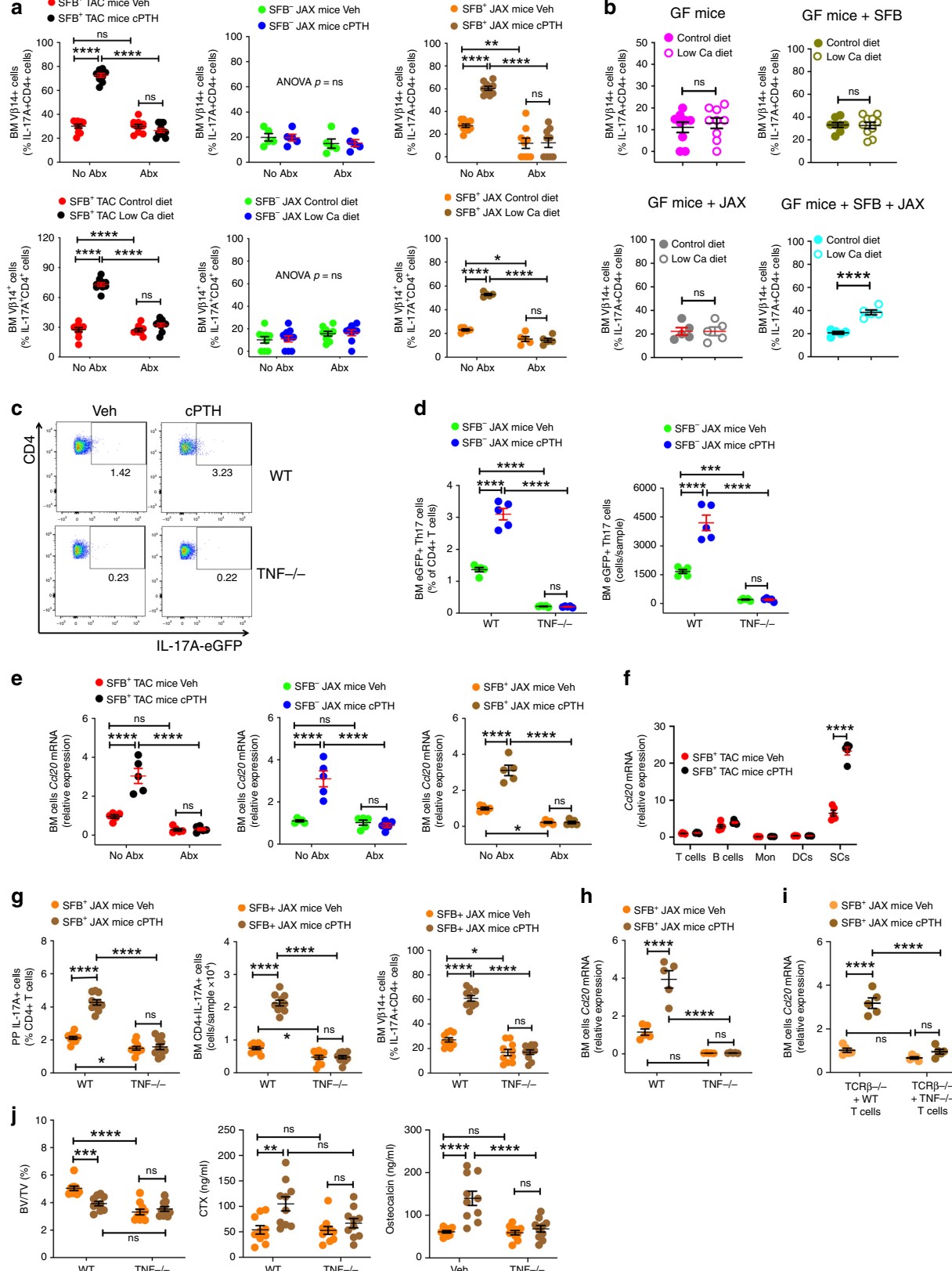

Data arguing against the possibility that PTH may have directly increased the differentiation of TNF+ T cells and Th17 cells in the BM are the findings that PTH did not expand these BM populations in the absence SFB in the intestinal microbiota. In addition, blockade of their egress from the gut, or their influx into the BM prevented their PTH-induced expansion within the BM. Further supporting data for an intestinal origin of PTH-induced BM Th17 cells was the finding of the expansion of intestinal derived Vβ14+ Th17 cells in the BM of cPTH treated mice, and the finding of increased tropism for the BM of exogenous Th17

**Fig. 8 TNF is sufficient for cPTH and low calcium diet to upregulate CCL20 levels in the BM and recruit Th17 cells to the BM. a** Relative frequency of BM Vβ14 + Th17 cells in SFB⁺ TAC and JAX mice treated with cPTH or low calcium, diet with antibiotics (Abx) or without antibiotics (No Abx), $n = 5$–10 mice per group. **b** Relative frequency of BM Vβ14 + Th17 cells in GF mice, SFB monoassociated GF mice, and GF mice colonized with JAX mice microbiota ±SFB treated with control diet or low calcium diet, $n = 5$–10 mice per group. **c, d** Representative FACS plot and frequency of eGFP + Th17 cells in the BM of WT and Tnf−/− mice treated with Veh or cPTH and previously subjected to adoptive transfer of IL-17A-eGFP⁺ cells, $n = 5$ mice per group. **e** BM cells transcript levels of Ccl20 in SFB⁺ and SFB⁻ TAC and JAX mice treated with and without Abx and cPTH, $n = 5$ mice per group. **f** Transcript levels of Ccl20 in purified BM T cells, B cells, monocytes (Mon), dendritic cells (DCs) and stromal cells (SCs), $n = 5$ mice per group. **g** Frequency of PP and BM Th17 cells, and BM Vβ14 + Th17 cells in WT and Tnf−/− mice, $n = 10$ mice per group. **h** BM cells Ccl20 transcripts in WT and Tnf−/− mice, $n = 5$ mice per group. **i** BM cells Ccl20 transcripts in Tcrb−/− mice reconstituted with WT T cells or Tnf−/− T cells, $n = 5$ mice per group. **j** BV/TV, serum CTX levels and serum osteocalcin levels in WT and Tnf−/− mice, $n = 10$ mice per group. In panels F-I SFB⁺ JAX mice were treated with Veh or cPTH. Data are expressed as Mean ± SEM. All data were normally distributed according to the Shapiro-Wilk normality test. Data in **a**, **d**–**j** were analyzed by two-way analysis-of-variance and post hoc tests applying the Bonferroni correction for multiple comparisons. Data in panel b were analyzed by unpaired t-tests. *$p < 0.05$, **$p < 0.01$, ***$p < 0.001$, and ****$p < 0.0001$ compared to the indicated group. Source data are provided as a Source Data file.

cells in response to cPTH. Thus, our findings identified the intestine as a proximal target relevant for the skeletal effects of PTH.

The findings that cPTH and low calcium diet increased BM Tnf and Il17a transcript levels were consistent with the hypothesis that TNF⁺ T cells and Th17 were first produced in the gut in response to the microbiota, and then homed to the BM where they contribute to increase the overall levels of BM TNF and IL-17A. However, we cannot conclusively exclude the possibility that PTH may have increased the production of TNF and IL-17A by resident BM lineages which were activated by microbial metabolites produced in the gut in response to PTH, which then diffused to the BM[48]. An unexpected finding of this study was that the deletion of CXCR3 blocked not only the PTH-induced influx of TNF⁺ T cells in the BM, but also the expansion of these cells in PPs. Since the microbiota induces the differentiation of helper T cells in the lamina propria, these findings may indicate that CXCR3 not only mediates the migration of TNF⁺ T cells from the gut to the BM, but also the transfer of these cells from the lamina propria to PPs. This hypothesis is consistent with the capacity of CXCR3 to mediate T cell migration to the intestine[49].

Direct PTH signaling within osteoblast and osteocytes that induces RANKL production by these cells contributes to PTH-induced bone loss[10,50]. However, T cells are also targeted by PTH, and play a role in PTH-induced bone loss by upregulating the capacity of osteocytes and osteoblasts to release RANKL in response to PTH[16,18,22]. Our findings point to an extraskeletal factor, namely the gut microbiome as a required determinant of the skeletal effects of continuous overproduction or infusion of PTH. Our data show a direct functional relationship between SFB induced intestinal Th17 cells, and Th17 cell driven skeletal damage.

The discoveries reported in this manuscript have significant clinical implications because the presence or absence of human equivalents of a Th17 cell-inducing microbiota in patients affected by primary or secondary hyperparathyroidism may determine the clinical manifestation of these conditions. For example, whether patients with primary hyperparathyroidism develop the classical form of the disease with hypercalcemia and bone loss, the normocalcemic variant with skeletal manifestations, or the normocalcemic variant without skeletal manifestations[2,7], might be dependent on the presence or the absence of one or more Th17 cell-inducing SFB equivalents in their gut flora. Should this be the case, stool microbiome sequencing would be an informative biomarker of primary hyperparathyroidism subtype, and antibiotics might be repurposed to prevent the skeletal manifestation of hyperparathyroidism. Moreover, the finding that blockade of Th17 cell intestinal egress or Th17 cell influx in the BM prevented the bone loss induced by PTH

provides proof of principle that pharmacological modulation of Th17 cell trafficking may represent a therapeutic strategy for hyperparathyroidism.

## Methods

This study was performed in compliance with all relevant ethical regulations for animal testing and research. All treatments and surgical procedures were approved by the Institutional Animal Care and Use Committee (IACUC) of Emory University.

**Mice.** All in vivo experiments were carried out in 16-week-old female mice. We utilized conventionally raised C57BL/6 mice from Taconic biosciences (Rensselaer, NY) and conventionally raised C57BL/6 mice, Tnf−/− mice (B6.129S6-Tnf<tm1Gkl>/J),TCRβ−/− mice (B6.129P2-Tcrbtm1Mom/J), Il17a-eGFP knock in mice (C57BL/6-il17a<tm1Bcgen>/J) and Cxcr3−/− mice (B6.129P2-Cxcr3<tm1Dgen>/J) from The Jackson Laboratory (Bar Harbor, ME). We also used SFB⁺ JAX mice that were generated by carrying out a fecal microbiome transfer (FMT) by oral gavage into SFB⁻ JAX mice with a liquid suspension of fecal pellets collected from SFB⁺ TAC mice. TAC mice were verified by us to be colonized by SFB (and JAX mice were verified to be SFB⁻) by fecal DNA extraction using QIAamp DNA Stool Mini Kit (QIAGEN) and subsequent quantitative PCR (qPCR) using established protocol that used primers that are specific for the SFB 16S rDNA gene[51]. Sequences of the primers are listed in supplementary table 1. All conventionally raised mice entering Emory University were shipped to the same room in the same vivarium within the Whitehead Biomedical Research Building. All conventionally raised mice were maintained under general housing environment. All mice were acclimatized within our facility for at least 3 days before experimentation.

Germ-free (GF) C57BL/6 mice were generated within the Emory Gnotobiotic Animal Core (EGAC). GF mice mono-colonized with SFB were generated by oral gavage of 3-week-old GF mice with fecal pellets that were collected from SFB mono-associated mice[52,53], which were a gift from Andrew T. Gewirtz (Georgia State University). Control mice for this experiment were generated by colonizing 3-week-old GF mice with fecal pellets collected from conventionally raised SFB⁻ JAX mice. In addition, further groups of mice were generated by colonization of 3-week-old GF mice by oral gavage of fecal pellet from conventionally raised SFB⁻ JAX mice and with fecal pellets collected from SFB mono-associated mice. The GF mice, the SFB mono-associated GF mice, the GF mice colonized with the SFB⁻ JAX microbiome, and the mice colonized with both the SFB⁻ JAX microbiota and SFB were housed in hermetically sealed Tecniplast ISOcage P Bioexclusion System within the EGAC for the duration of the experiment to ensure that no other bacteria enters the microbiome. All Mice were randomly assigned to experimental groups.

**In vivo cPTH treatment.** 80 μg/kg/day of hPTH1-34 (Bachem California Inc., Torrance, CA) or vehicle were infused in 16-week-old female mice by implanting ALZET osmotic pump model-1002 (DURECT corporation Cupertino, CA) with a delivery rate of 0.21 μl/h, as previously described[16–18,22].

**Mouse diet.** Mice treated with cPTH or vehicle were fed sterilized food (5V5R chow) made by LabDiet (St. Louis MO) that contains 0.99% calcium. Mice enrolled in the low calcium diet experiments were fed sterilized food made by MP Biomedicals (Solon, OH) containing either 0.01% calcium or 1.75% calcium for 28 days. All mice received autoclaved water ad libitum.

**Depletion of gut commensal microflora.** A cocktail of antibiotics (1 mg/mL Ampicillin, 0.5 mg/mL Vancomycin, 1 mg/mL Neomycin Sulfate, and 1 mg/mL Metronidazole Benzoate/Metronidazole) was prepared as described in Rakoff-

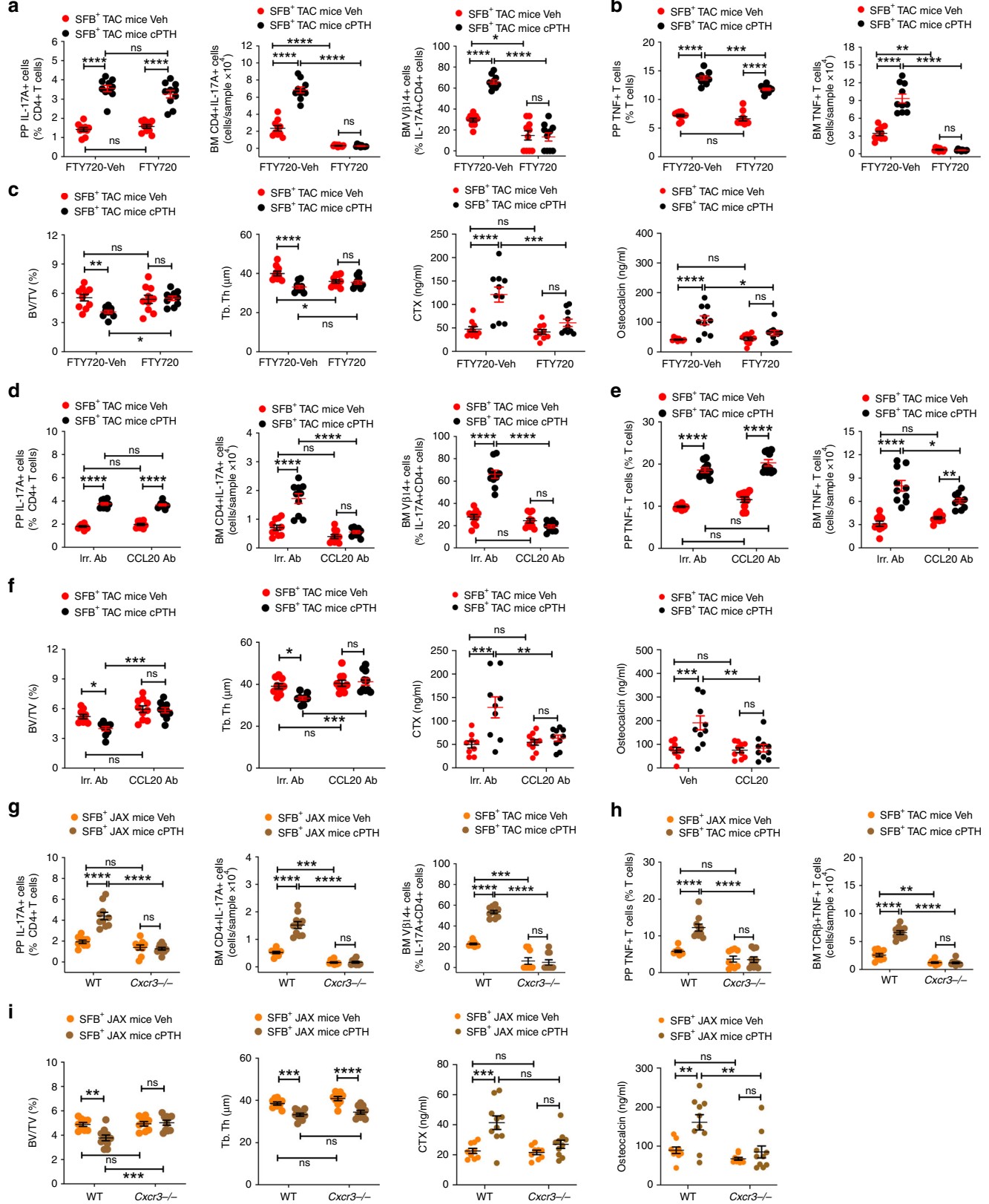

Nahoum et al.[54], and included in the drinking water of mice starting at 2 weeks before initiation of cPTH treatment or low calcium diet feeding. Antibiotic water was prepared freshly and changed twice a week.

**PP excision end preparation of single cell suspension**. PP cell isolation was performed as described[55]. Briefly, the small intestine was removed and flushed of fecal content. PPs were excised and collected in 1 ml cooled RPMI1640. PPs were

dissociated using the plunger of a 2.5 ml syringe and gently forced through a 70 μm cell strainer placed over a 50 ml tube. A single cell suspension was used for analysis by flow cytometry.

**Stromal cell purification**. BM cells from long bones were cultured for 7 days in α-MEM medium containing 10% FBS, 100 mg/mL of penicillin, and 100 IU/mL of streptomycin, to allow the proliferation of stromal cells (SCs). After removing non

**Fig. 9 Blockade of T cell egress from the intestine or of T cell influx in the BM prevent cPTH induced bone loss. a** Effects of cPTH on the number of PP and BM Th17 cells and BM Vβ14 + Th17 cells in SFB+ TAC mice treated with FTY720, n = 10 mice per group. **b** Effects of cPTH on the number of PP and BM TNF+ T cells in SFB+ TAC mice treated with FTY720, n = 10 mice per group. **c** Effects of cPTH on BV/TV, Tb.Th, serum CTX levels and serum osteocalcin levels in SFB+ TAC mice treated with FTY720, n = 10 mice per group. **d** Effects of cPTH on the number of PP and BM Th17 cells and BM Vβ14 + Th17 cells in SFB+ TAC mice treated with anti-CCL20 Ab, n = 10 mice per group. **e** Effects of cPTH on the number of PP and BM TNF+ T cells in SFB+ TAC mice treated with anti-CCL20 Ab, n = 10 mice per group. **f** Effects of cPTH on BV/TV, Tb.Th, serum CTX levels and serum osteocalcin levels in SFB+ TAC mice treated with anti-CCL20 Ab, n = 9–10 mice per group. **g** Effects of cPTH on the number of PP and BM Th17 cells and BM Vβ14 + Th17 cells in SFB+ JAX Cxcr3−/− mice and WT littermates, n = 9–10 mice per group. **h** Effects of cPTH on the number of PP and BM TNF+ T cells in SFB+ JAX Cxcr3−/− mice and WT littermates, n = 9–10 mice per group. **i**. Effects of cPTH on BV/TV, Tb.Th, serum CTX levels and serum osteocalcin levels in in SFB+ JAX Cxcr3−/− mice and WT littermates, n = 9–10 mice per group. Data are expressed as Mean ± SEM. All data were normally distributed according to the Shapiro-Wilk normality test and analyzed by two-way analysis-of-variance and post hoc tests applying the Bonferroni correction for multiple comparisons. *$p < 0.05$, **$p < 0.01$, ***$p < 0.001$, and ****$p < 0.0001$ compared to the indicated group. Source data are provided as a Source Data file.

---

adherent cells, adherent macrophages were eliminated by positive immunoselection using anti-CD11c MACS Microbeads (Miltenyi Biotech). The remaining adherent cells were defined as SCs as they express alkaline phosphatase (ALP), type-I collagen, and Runx2, and have the capacity to form mineralization nodules when further cultured under mineralizing conditions.

**Purification of T and B cells, monocytes, and DCs.** These lineages were purified from murine BM by positive immuno-magnetic selection:

PE anti-mouse TCR β antibody (clone H57-597) (Biolegend) and EasySep Mouse PE Positive Selection Kit (StemCell Technologies) for T cells, PE anti-mouse CD19 antibody (clone 6D5) (Biolegend) and EasySep Mouse PE Positive Selection Kit (StemCell Technologies) for B cells, anti-CD11b MicroBeads (Miltenyi Biotech) for monocytes, and MojoSort Mouse CD11c Nanobeads (Biolegend) for dendritic cells.

**T cell transfers.** Il17a-GFP knock-in mice (C57BL/6-IL17atm1Bcgen/J) were purchased from Jackson Laboratory. These mice express enhanced GFP (eGFP) as a marker of IL17a activity. Splenic naïve CD4+ cells (CD4+ CD44lo CD62Lhi) were isolated using EasySep Mouse Naïve CD4+ T Cell Isolation Kit (StemCell Technologies) from IL-17A-eGFP mice and cultured in Th17 polarizing condition for 4 days using Mouse Th17 Cell Differentiation Kit (R&D). eGFP+ CD4+ live T cells were FACS sorted by a FACSAria II (BD Biocsiences) and injected (1 × 10⁶ cells per mouse) IV into SFB− JAX WT and Tnf−/− recipient mice previously treated with vehicle or cPTH for 6 days. 3 days later eGFP+ CD4+ T cells in BM of recipient mice were analyzed by flow cytometry.

WT and Tnf −/− spleen T cells were purified by negative immunoselection using MACS Pan T cell isolation kit (Miltenyi Biotech). These cells were injected (5 × 10⁶ cells per mouse) I.V. into TCRβ−/− recipient mice 2 weeks before treatment. Successful T cell engraftment was confirmed by flow cytometry of the spleens of the recipient mice harvested at sacrifice.

**Pharmacological treatments.** FTY720 was added to the drinking water at 5 µg/mL as described in Krebs et al.[34]. Water containing FTY720 was changed weekly. FTY720 treatment was initiated one day before cPTH and continued until sacrifice. Rat anti-CCL20-antibody (clone 114908, R&D Systems) or isotype control (clone 43414, R&D Systems) were injected (i.p.) at 50 µg per mouse one day before initiation of cPTH treatment and every other day thereafter until sacrifice.

**µCT measurements.** µCT scanning and analysis of the distal femur was performed as reported previously[18,56–58] using a Scanco µCT-40 scanner (Scanco Medical, Bassersdorf, Switzerland). Femoral trabecular and cortical bone regions were evaluated using isotropic 12-µm voxels. For the femoral trabecular region, we analyzed 70 slices starting eight slices below the distal growth plate. Femoral cortical bone was assessed using 80 continuous CT slides located at the femoral midshaft. X-ray tube potential was 70 kVp, and integration time was 200 ms. We used the thresholding approach described by Bouxsein et al.[59], which is recommended by Scanco, the mCT-40 manufacturer, and involves a visual inspection and comparison of preview and slice-wise gray scale 2D images. The same threshold value was used for all measurements.

**Quantitative bone histomorphometry.** The measurements, terminology and units used for histomorphometric analysis, were those recommended by the Nomenclature Committee of the American Society of Bone and Mineral Research[60]. Non-consecutive longitudinal sections of the femur were prepared and analyzed as described previously[18]. Mice were injected subcutaneously with calcein at day 10 and day 3 before sacrifice. Non-consecutive longitudinal sections (5 µm thick) were cut from methyl methacrylate plastic-embedded blocks along the frontal plane using a Leica RM2155 microtome and were stained with Goldner's trichrome stain for the static measurements. Additional sections were cut at 10 µm and left unstained for dynamic (fluorescent) measurements. Measurements were obtained

in an area of cancellous bone that measured ~2.5 mm² and contained only secondary spongiosa, which was located 0.5–2.5 mm proximal to the epiphyseal growth cartilage of the femurs. Measurements of single-labeled and double-labeled fluorescent surfaces, and interlabel width were made in the same region of interest using unstained sections. Analysis was done using the Bioquant Image Analysis System (R&M Biometrics). Mineral Apposition Rate (MAR) and Bone Formation Rate (BFR) were calculated by the software by applying the interlabel period.

**Markers of bone turnover.** Serum CTX and osteocalcin were measured by rodent specific ELISA assays (Immunodiagnostic Systems, Gaithersburg, MD).

**Flow cytometry.** Flow cytometry was performed on a LSR II system (BD Biosciences) and data were analyzed using FlowJo software (Tree Star, Inc., Ashland, OR). For cell surface staining: cells were stained with anti-mouse purified CD16/32 (clone 93), BV 510-CD45 (clone 30-F11), BV 421-TCRβ (clone H57-597), AF 700-CD3 (clone 17A2), PerCP/Cy5.5-CD4 (clone RM4-5), BV 711-CD8 (clone 53-6.7), AF 488-CD3ε (clone 145-2C11), BV 421-TCR γ/δ (clone GL3), PerCP/Cy5.5-F4/80 (clone BM8), BV 650-CD11b (clone M1/70), APC-Ly-6G (clone 1A8) (Biolegend) and FITC-Vβ T cell receptor (BD Biosciences). The live cells were discriminated by Zombie NIR Fixable Viability Kit (Biolegend) or LIVE/DEAD Fixable Yellow Dead Cell Stain Kit (ThermoFisher). For intracellular staining, cells were incubated with cell activation cocktail (Biolegend) in the presence of Monensin Solution at 37 °C for 12 h. Anti-mouse PE-IL-17A (clone eBio17B7), APC-Foxp3 (clone FJK-16s) (ThermoFisher), APC-TNF (clone MP6-XT22) (BD Biosciences), AF 488-IFNγ and PE/Dazzle 594-IL-4 antibodies (Biolegend) were added after cell fixation and permeabilization with Intracellular Fixation & Permeabilization Buffer Set (ThermoFisher).

**Real-time RT-PCR and primers.** Total RNA was isolated using TRIzol reagent (ThermoFisher Scientific) and DNase Max kit (QIAGEN) according to the manufacturer's directions. For all RNA samples, cDNA was synthesized with random hexamer primers (Roche) and AMV reverse transcriptase (Roche). The expression levels of murine Il17a, Tnf, and Tgfb were measured in whole BM cells by real-time PCR. Changes in relative gene expression between vehicle and cPTH groups or control and low Ca diet groups were calculated using the 2−ΔΔCT method with normalization to 18S rRNA. The primers used are provided in Supplementary Table 1.

**RNA isolation from small intestine.** A 10-mm intestinal piece of each mouse was collected at sacrifice and was kept at −80°C for storage. Tissue was disrupted and homogenized using RNase-free stainless steel beads in the Bullet Blender Strom 24 blender (Next Advance), RNA was extracted using TRIzol reagent (ThermoFisher Scientific) and DNase Max kit (QIAGEN). The expression levels of murine Il1β, Il6, Il17a, Tnf, and Tgfb were measured in SI cells by real-time PCR.

**Quantification and statistical analysis.** All data were normally distributed according to the Shapiro-Wilk normality test. Data were analyzed by unpaired two tailed t tests, one-way ANOVA, or two-way ANOVA, as appropriate. This analysis included the main effects for animal strain and treatment plus the statistical interaction between animal strain and treatment. When the statistical interaction was statistically significant ($p < 0.05$) or suggestive of an important interaction, then t tests were used to compare the differences between the treatment means for each animal strain, applying the Bonferroni correction for multiple comparisons. Data that were not normally distributed (as tested by Shapiro-Wilk normality test) were analyzed by Kruskal-Wallis non-parametric tests.

**Reporting summary.** Further information on research design is available in the Nature Research Reporting Summary linked to this article.

## Data availability

The datasets generated during and/or analyzed during the current study are available from the corresponding author on reasonable request.

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

## Acknowledgements

This study was supported by grants from the National Institutes of Health (RP: DK112946, DK108842, and RR028009; RMJ: DK098391; MNW: AG062334, AR068157 and AR070091). MNW was also supported by a grant from the Biomedical Laboratory Research & Development Service of the VA Office of Research and Development (5I01BX000105).

## Author contributions

R.M.J., T.L.D., M.N.W., A.G., and R.P. designed the studies. M.Y. developed and optimized protocols, M.Y., A.M.T., J.Y.L., and J.A. performed the research and analyzed the data. R.M.J., T.L.D., M.N.W., and R.P. wrote the paper.

## Competing interests

The authors declare no competing interests.
