## [Peer Review File · Nature Communications]

Reviewers' Comments:

Reviewer #1:

Remarks to the Author:

The paper by Yu et al. shows that primary and secondary PTH-induced trabecular bone loss in mice is dependent on intestinal microbiota, or more specifically on the presence of SFB in the context of polymicrobial intestinal microbiota. The phenomenon is demonstrated in well-controlled experiments with SFB- and SFB+ WT mice as well as SFB- WT mice reconstituted with SFB. Low-Ca diet induced bone loss is further demonstrated with GF mice reconstituted with SFB alone, SFB-feces or SFB+ feces. Mechanistically, SFB+ microbiota-dependent influx of Th17 cells into the bone marrow is demonstrated, which requires TNF and is prevented by treating the mice with FTY720, anti-CCL20 or in CXC3R-/- mice. The paper provides novel insight into the mechanism of PTH-induced bone loss. The experimental approach is very methodical and results are well documented.

Major comments

- While the data are certainly consistent with a role for TNF+ T cells in up-regulating CCL20 in the bone marrow and recruiting Th17 cells etc. this is not formally demonstrated in the paper (it would require experiments with conditional T cell-specific TNF deficient mice). Beware of overinterpreting the data in the discussion.

Minor comments

- Consider including a figure 1 that shows the experimental design visually.
- If ANOVA is used, post hoc testing should only be applied if the overall p value is significant. Showing 4 "ns" comparisons in a graph (e.g. Fig. 1B middle graph) is therefore inappropriate as the overall p value is presumably also not significant.
- Please show a scale bar in the histology sections (Fig. 1 D, E). Original magnification values are of limited value in the digital age.
- The figures are generally very dense with many graphs. I suggest to include panel headings, e.g. "B trabecular bone BV/TV ". Readers' necks will thank the authors for that. Because of the consistent color coding (nice job !), the color legends do not need to be shown above every graph (Fig. 5 and 6). Replacing these redundant mini legends with panel titles would improve readability. Consider reducing the color palette, for instance, one could show control and low Ca diet as full and empty circles of a single color.

Reviewer #2:

Remarks to the Author:

The manuscript of Yu et al. is a novel and interesting paper about gut-bone crosstalk mediated by microbiota in primary and secondary hyperparathyroidism. In the well-conducted study, the authors demonstrate in different mouse models of primary and secondary hyperparathyroidism that PTH-induced bone loss is dependent on segmented filamentous bacteria (SFB)-induced expansion of Th17 cells. With the help of state-of-the art methods and appropriate animal models, they provide a model how Th17 cells are recruited from the intestine to the bone marrow through a sphingosine-1-phosphate (S1P) receptor 1 (S1P1R) dependent mechanism that is guided by TNF+ T cell dependent induction of CCL20 in the bone marrow.

The work is important and sheds new light on the functional role of microbiota in hyperparathyroidism caused bone loss.

The data are clear and convincing and they are presented in a straightforward way. I think the overall quality of the paper is suitable for Nature Communication. I just have few comments.

1) The authors show that cPTH treatment increases the expression of TGF β in the small intestine. As TGF β is also essential for Treg differentiation, it would be interesting to see if and how cPTH treatment directly affects the population of Treg cells. In Figure S8, the authors only report on the effects of low calcium diet on Treg cell population.

2) Along this line: How are other immune cells such as $\gamma\delta$ T cells, which also produce IL-17, and neutrophils are affected by cPTH treatment in the intestine?

3) cPTH treatment increased CCL20 mRNA expression in bone marrow cells. Can the authors specify which immune cells express CCL20 upon cPTH treatment? Are the TNF+ T cells that migrate to the bone marrow responsible for the increase in CCL20 expression?

4) Can the authors add a graphical scheme of the proposed working model?

Reply to Reviewer 1

The paper by Yu et al. shows that primary and secondary PTH-induced trabecular bone loss in mice is dependent on intestinal microbiota, or more specifically on the presence of SFB in the context of polymicrobial intestinal microbiota. The phenomenon is demonstrated in well-controlled experiments with SFB- and SFB+ WT mice as well as SFB- WT mice reconstituted with SFB. Low-Ca diet induced bone loss is further demonstrated with GF mice reconstituted with SFB alone, SFB- feces or SFB+ feces. Mechanistically, SFB+ microbiota-dependent influx of Th17 cells into the bone marrow is demonstrated, which requires TNF and is prevented by treating the mice with FTY720, anti-CCL20 or in CXC3R^{-/-} mice. The paper provides novel insight into the mechanism of PTH-induced bone loss. The experimental approach is very methodical, and results are well documented.

Reply. We are grateful to Reviewer 1 for the nice comments about the paper.

Major comments

- While the data are certainly consistent with a role for TNF⁺ T cells in up-regulating CCL20 in the bone marrow and recruiting Th17 cells etc. this is not formally demonstrated in the paper (it would require experiments with conditional T cell-specific TNF deficient mice). Beware of overinterpreting the data in the discussion.

Figure 8i. BM cells CCL20 transcripts in TCRβ^{-/-} mice reconstituted with WT T cells or TNF^{-/-} T cells. *= $p < 0.05$, **= $p < 0.01$, ***= $p < 0.001$ and ****= $p < 0.0001$ compared to the indicated group.

Reply: We agree with Reviewer 1. The data presented in the original manuscript obtained using TNF^{-/-} mice show that TNF is required for cPTH to upregulate CCL20 in the bone marrow (BM). However, the manuscript did not provide evidence that TNF⁺ T cells are responsible for this effect. Thanks to the availability of samples from another study, we are now able to show the results of an experiments in which T cells from TNF^{-/-} mice were adoptively transferred into TCRβ^{-/-} mice, a strain devoid of αβ T cells, to generate chimeric mice specifically lacking the production of TNF by T cells. In revised figure 8i we show that cPTH upregulates CCL20 transcripts in mice with WT T cells, but not in those lacking TNF productions by T cells. We feel that this new evidence resolves the issue of overinterpretation of the data in the discussion noted by Reviewer 1.

Minor comments

- Consider including a figure 1 that shows the experimental design visually.

Reply: Done. The experimental design is now visually shown in Fig S1.

- If ANOVA is used, post hoc testing should only be applied if the overall p value is significant. Showing 4 "ns" comparisons in a graph (e.g. Fig. 1B middle graph) is therefore inappropriate as the overall p value is presumably also not significant.

Reply: Reviewer 1 is correct. When appropriate, in the revised manuscript we have removed the “ns” relative to post-hoc tests and replaced with a “ns” for the overall ANOVA.

- Please show a scale bar in the histology sections (Fig. 1 D, E). Original magnification values are of limited value in the digital age.

Reply: Done

- The figures are generally very dense with many graphs. I suggest to include panel headings, e.g. “B trabecular bone BV/TV ”. Readers’ necks will thank the authors for that. Because of the consistent color coding (nice job !), the color legends do not need to be shown above every graph (Fig. 5 and 6). Replacing these redundant mini legends with panel titles would improve readability. Consider reducing the color palette, for instance, one could show control and low Ca diet as full and empty circles of a single color.

Reply: We agree that readers’ neck health is important! Nature Communication style rules prevent us from adding text to the letter defining each panel, but we have added horizontal labels defining the measured indices in those figures where we had the space to do so. In order to show these labels in as many figures as possible we have also rearranged some of the figures and moved some data from supplemental figures to main figures. We have also reduced the color palette. We would prefer to keep the color legend in the original figure 5 and 6 to allow readers that want to look only at one figure to understand the groups without having to go back to earlier figures.

Reply to Reviewer #2

The manuscript of Yu et al. is a novel and interesting paper about gut-bone crosstalk mediated by microbiota in primary and secondary hyperparathyroidism. In the well-conducted study, the authors demonstrate in different mouse models of primary and secondary hyperparathyroidism that PTH-induced bone loss is dependent on segmented filamentous bacteria (SFB)-induced expansion of Th17 cells. With the help of state-of-the-art methods and appropriate animal models, they provide a model how Th17 cells are recruited from the intestine to the bone marrow through a sphingosine-1-phosphate (S1P) receptor 1 (S1P1R) dependent mechanism that is guided by TNF+ T cell dependent induction of CCL20 in the bone marrow.

The work is important and sheds new light on the functional role of microbiota in hyperparathyroidism caused bone loss.

The data are clear and convincing, and they are presented in a straightforward way. I think the overall quality of the paper is suitable for Nature Communication. I just have few comments.

Reply: We are very grateful for the nice comments for Reviewer 2.

1) The authors show that cPTH treatment increases the expression of TGF β in the small intestine. As TGF β is also essential for Treg differentiation, it would be interesting to see if and how cPTH treatment directly affects the population of Treg cells. In Figure S8, the authors only report on the effects of low calcium diet on Treg cell population.

Reply: This is an interesting point. In revised figure S7a,b we show that neither cPTH nor low Ca diet expand Tregs in the small intestine.

Figure S7. cPTH and low calcium diet do not affect the frequency of PP and BM Foxp3+ T cells. a. PP and BM Foxp3+ T cells, in SFB⁺ JAX mice treated with vehicle or cPTH for 2 weeks. b. PP and BM Foxp3+ T cells, in SFB⁺ JAX mice treated

2) Along this line: How are other immune cells such as $\gamma\delta$ T cells, which also produce IL-17, and neutrophils are affected by cPTH treatment in the intestine?

Reply: In revised figure S5 we show that cPTH increased the relative frequency of PP IL-17A⁺ CD4⁺ T cells (Th17 cells) but not of PP IL-17A⁺ $\gamma\delta$ T cells and (fig S5a). The frequency of neutrophils in PPs was too low to reliably assess their production of IL-17. Moreover, cPTH did not affect the frequency of PP neutrophils (fig S5b).

Figure S5. Effects of cPTH treatment on the relative frequency of PP IL-17A⁺ $\gamma\delta$ T cells, Th17 cells and neutrophils. a. Relative frequency of PP IL-17A⁺ $\gamma\delta$ T cells and PP IL-17A⁺ CD4⁺ T cells expressed as percentage of total T cells (CD3 ϵ ⁺ cells). b. Relative frequency of PP granulocytes.

3) cPTH treatment increased CCL20 mRNA expression in bone marrow cells. Can the authors specify which immune cells express CCL20 upon cPTH treatment? Are the TNF+ T cells that migrate to the bone marrow responsible for the increase in CCL20 expression?

Reply: In revised figure 8f we provide new data showing that cPTH upregulates CCL20 mRNA expression by bone marrow stromal cells (SCs), but not from immune cells. This is consistent with previously published data.

Thanks to the availability of samples from another study, we are now able to show the results of an experiments in which T cells from TNF^{-/-} mice were adoptively transferred into TCRβ^{-/-} mice, a strain devoid of αβ T cells, to generate chimeric mice specifically lacking the production of TNF by T cells. In revised figure 8i we show that cPTH upregulates CCL20 transcripts in control mice but not in those lacking T cell TNF production.

Figure 8f. Transcript levels of CCL20 in purified BM T cells, B cells, monocytes (Mon), dendritic cells (DCs) and stromal cells (SCs). ****= p<0.0001 compared to the indicated group.

Figure 8i. BM cells CCL20 transcripts in TCRβ^{-/-} mice reconstituted with WT T cells or TNF^{-/-} T cells. * = p<0.05, ** = p<0.01, *** = p<0.001 and **** = p<0.0001 compared to the indicated group.

4) Can the authors add a graphical scheme of the proposed working model?

Reply: As per Nature Communication editorial policy, we provide a graphical scheme of the working model as graphic abstract.

Reviewers' Comments:

Reviewer #1:

Remarks to the Author:

My comments have been adequately addressed in the revised manuscript.

Reviewer #2:

Remarks to the Author:

My comments have been adequately addressed.